# Material Technologies for Improved Diabetic Foot Ulcer (DFU) Treatment: A Questionnaire Study of Healthcare Professionals’ Needs

**DOI:** 10.3390/biomedicines12112483

**Published:** 2024-10-29

**Authors:** Marian Gabriela Vargas Guerrero, Lieve Vonken, Erwin Peters, Jimmy Lucchesi, Jacobus J. C. Arts

**Affiliations:** 1Department of Orthopaedic Surgery, Maastricht University Medical Centre (MUMC+), 6229 HX Maastricht, The Netherlands; marian.vargasguerrero@maastrichtuniversity.nl (M.G.V.G.);; 2Laboratory for Experimental Orthopaedics, Faculty of Health, Medicine & Life Sciences, Maastricht University, 6229 ER Maastricht, The Netherlands; 3Department of Health Promotion, Faculty of Health, Medicine & Life Sciences, Maastricht University, 6229 HA Maastricht, The Netherlands; 4Bonalive Biomaterials, 20750 Turku, Finland; 5Department of Orthopaedic Biomechanics, Faculty of Biomedical Engineering, Eindhoven University of Technology, 5600 MB Eindhoven, The Netherlands

**Keywords:** diabetic foot ulcers, osteomyelitis, wound care, wound dressings, material technologies, user needs

## Abstract

**Background/Objectives:** Diabetic foot ulcers (DFUs) are a common and severe complication of diabetic patients, with significant global prevalence and associated health burdens, including high recurrence rates, lower-limb amputations, and substantial associated economic costs. This study aimed to understand the user needs of healthcare professionals treating diabetic foot ulcers for newly developed material technologies. **Methods:** An open-ended questionnaire was used to identify user needs, identify the limitations of current treatments, and determine the specific requirements for ideal treatment. This information was used to develop a list of key considerations for creating innovative material technologies to improve diabetic wound treatment results. **Results:** Most respondents indicated that they followed published treatment guidelines for DFUs but noted that treatment often required a case-specific approach. Antibiotics and surgical debridement were commonly used for infection control. The participants showed a strong preference for wound dressings with lasting antibacterial properties. Respondents identified ideal properties for new products, including ease of use, enhanced antibacterial properties, affordability, and targeted biological activity. The respondents also highlighted the importance of a holistic approach to DFU management, integrating product development with comprehensive care strategies and patient education. **Conclusions:** This study highlights the complexity of DFU care, emphasizing that no single product can address all treatment needs. Future materials could focus on combination therapies and specific use cases. Additionally, understanding global variations in treatment practices and educating users on the proper application of newly developed material technologies is crucial for improving the management of DFUs and patient outcomes.

## 1. Introduction

Diabetic foot ulcers represent a significant health burden, characterized by high incidences, recurrence rates, and substantial associated economic costs. Diabetic foot ulcers have an annual incidence rate of 2% in the general diabetic population [1]. However, individuals with established neuropathy are at an even higher risk, experiencing ulceration rates between 5.0% and 7.5% [2]. Throughout their lifetime, diabetic patients face a 19 to 34% risk of developing a foot ulcer. This translates into 9.1 million to 26.1 million diabetic patients with foot ulcers per year [2]. Following the initial ulceration, morbidity remains high. Recurrence rates reach 65% within 3–5 years, and patients experience a 20% lifetime risk of lower-limb amputation alongside a 50–70% mortality rate within 5 years [3,4,5,6]. The consequences of DFUs extend beyond patient well-being, imposing a significant economic burden on healthcare systems. In Europe, for instance, annual costs per DFU patient range from EUR 7000 to EUR 13,500, with substantial variations attributable to healthcare systems, treatment approaches, and study methodologies [7]. However, these costs pale compared to the lifetime costs associated with amputation, estimated at USD 350,465–USD 509,2645 per patient [5,8]. DFUs present a global prevalence of 6.3% [9] and are caused by a combination of neuropathy (nerve damage), trauma, peripheral artery disease (PAD), excessive pressure on the foot [10], and chronic kidney disease [11]. DFUs are susceptible to infection [12,13], with over 50% of ulcerations developing an infection [14]. This high rate of infection is attributed to compromised immune function [15], poor circulation [16], edema, neuropathy, and prolonged inflammation, all of which may be present in diabetic patients [17].

The specific classification of a DFU is crucial for providing appropriate wound care and therapeutic interventions [10]. DFUs can be classified as neuropathic [18], ischemic [19], or neuroischemic [20]. Neuropathic ulcers develop due to nerve damage, leading to sensory loss, callus formation, and increased pressure on the foot, often resulting in deep, painless ulcers on the plantar surface [18]. Ischemic ulcers arise from reduced blood flow caused by peripheral artery disease (PAD), presenting as painful, pale wounds with poor healing potential [19]. Neuroischemic ulcers, the most prevalent type of DFUs, combine features of both neuropathic and ischemic ulcers, occurring in individuals with both nerve damage and reduced blood flow [20,21]. A study performed in Catalonian primary care centers shows that the prevalence of neuroischemic ulcers among DFUs was 44.1% while 20.3% were neuropathic, and 20.3% were ischemic [21].

The International Working Group on the Diabetic Foot (IWGDF) Guidelines recommends that DFU treatment should involve a multidisciplinary team to provide individualized care tailored to the complex needs of each patient [22]. This guideline recommends that treatment consists of extensive surgical debridement, antibiotic therapy [23], topical dressings, wound decompression, vascular assessment, and glycemic control [24,25]. Dressings play a fundamental role in the quality and speed of wound healing [26]; however, dressings suitable for chronic wounds, such as diabetic foot ulcers are not currently available [27]. DFUs remain a complex therapeutic challenge despite recent advances in diabetes treatment and wound dressings [16,28,29]. Hyperglycemia in diabetes leads to inflammation, hypoxia, and immune dysfunction, increasing infection risk due to delayed leukocyte migration [30]. Fibroblasts exhibit impaired differentiation and cytokine release, hindering adaptive wound healing [31]. Epithelial cells such as keratinocytes have their migration and proliferation inhibited, delaying re-epithelialization and prolonging wound closure [32]. Diabetes also impairs the signaling of vascular endothelial growth factor (VEGF), which is key in promoting angiogenesis; this affects the growth and function of endothelial cells which are essential for new blood vessel formation [33]. If the wound becomes infected, the healing process is further challenged [34]. The prolonged inflammation in DFUs delays wound healing; additionally, bacteria produce enzymes that break down newly formed tissue and extracellular matrix components [17,35]. These factors underscore the complexity of DFUs and the need for specialized treatment.

Traditional and modern dressings are readily available, although all of them present limitations in wound management [36]. Traditional dressings require frequent replacement due to inadequate exudate absorption, leading to potential tissue damage and increased patient discomfort [37]. Moreover, they tend to create a dry wound environment [38], hindering granulation tissue formation and delaying wound healing [36,37]. Modern wound dressings offer significant advancements in wound healing, including enhanced granulation tissue formation, reduced inflammation, and antibacterial properties [39,40]. However, while offering numerous benefits, they also present limitations. For instance, the moist environment they create can lead to maceration of surrounding skin if not managed properly [41]. Dressings such as foams do not allow us to visualize the wound without removal, and this makes clinical monitoring difficult [41]. Modern wound dressings are also more expensive than traditional dressings [27] and may require a secondary dressing to ensure adhesion to the wound bed. Many new materials used in advanced dressings require complex manufacturing techniques that limit widespread production and availability [42]. Storage can also be an issue, as some advanced dressings may have specific storage requirements or are more difficult to handle than traditional dressings [27]. Given the advantages and limitations of current wound dressings, we decided to explore the user needs of healthcare professionals who use advanced dressings to treat diabetic foot ulcers (DFUs). The poor clinical outcomes of diabetic ulcers in comparison to non-diabetic ulcers [2,43] made it clear that this investigation was necessary. This led us to create a user needs questionnaire to investigate what an ideal product for treating DFUs would resemble from the perspective of healthcare professionals directly involved in such treatments.

This exploratory study uses a questionnaire to ascertain what healthcare professionals and material scientists experienced in DFUs consider relevant when developing therapeutic material technologies and products.

First, we inquired about the current challenges, limitations, and unmet needs.Secondly, we inquired about what healthcare professionals consider an ideal treatment.Third, we created an overview of what user needs should be considered when developing innovative therapeutic material technologies to treat DFUs.

## 2. Methods

### 2.1. Research Design

The open-ended questionnaire was distributed from January 2023 to September 2023 to purposively sample healthcare professionals and material experts with experience in DFU treatment.

Questionnaire development

The first draft of the user needs questionnaire was developed (MV) based on a systematic literature review of emerging materials for DFU treatment, published by our group [44], and an existing user needs questionnaire by a product manager specializing in biomaterials. Subsequentially, material experts (n = 3) and clinicians (n = 2) were consulted to review and formulate the final version of the questionnaire.

The final questionnaire (Appendix A) included three sections and sixteen questions.

Section 1 (7 questions): addressed objective 1, which involved identifying current treatment options and shortcomings.Section 2 (7 questions): focused on objective 2, exploring the desired user needs of a treatment product for DFUs.Section 3 (2 questions): provided respondents with the opportunity to offer additional insights and clarify any points in the questionnaire.

### 2.2. Data Collection

The questionnaire was distributed by a QR code with a link to a Google Forms questionnaire, through the European Wound Management Association (EWMA), the 14th Pisa International Diabetic Foot Course in Pisa, the Oxford Bone Infection Conference (OBIC), and the Annual Meeting of the European Bone & Joint Infection Society (EBJIS). These channels ensured respondents had experience in the clinical wound treatment field. Respondents could complete the questionnaire on paper or online through a Google Form. Questionnaires filled out on paper were transcribed to the Google Form (MV).

### 2.3. Data Analysis

The questionnaire data were extracted to Microsoft Excel, and all answers were read by MV and JA to familiarize themselves with the data. Then, MV coded sections of the respondents’ answers using the hybrid approach to thematic analysis as described by Feredey and Muir-Cochrane [45]; answers were coded inductively as well as deductively. Qualitative content analysis was applied to further analyze reoccurring themes. The frequency of occurrence of these themes was reported. The quantitative data were supplemented with quotes from respondents. The codes used can be found in Appendix B.

## 3. Results

### 3.1. Sample Characteristics

The questionnaire received 29 responses. All respondents had experience in the treatment of DFUs or the development of DFU material technologies. Respondents most often worked in Europe (n = 24; Australia [n = 2], the United States [n = 2], or the Philippines [n = 1]). Most respondents worked in a hospital (n = 19), which was most often a university hospital (n = 12). The other respondents (n = 10) worked in peripheral clinics, as wound consultants, or were self-employed. Respondents worked in plastic surgery, orthopedics, dermatology, internal medicine, or the development of medical devices. Most professionals mentioned they were experts in wound healing (n = 15), and this includes physicians and nurses.

The results presented correspond to the analysis of the information from the user needs questionnaire, which was divided into three sections.

### 3.2. Section 1: Participant Background—Profiles of Commonly Treated Patients and Wounds Managed

#### 3.2.1. Medical and Demographic Characteristics of Patients Treated for Diabetic Wounds

The characteristics of patients treated by the healthcare professionals (respondents) are as follows:

**Age**: Sixteen respondents indicated that they generally treat patients above 60 years of age. Ten respondents did not specify an age or just mentioned that all age groups were difficult to treat for different reasons. Three respondents said that young people were more difficult to treat as they wanted to be more physically active.

**Medical Status**: Many patients treated by the respondents have additional health issues such as cardiovascular disease, obesity, neuropathy, and immune system impairments. These comorbidities significantly affect wound healing and treatment approaches. Out of the 29 respondents, 25 answered this question. In total, 32% of the respondents (n = 8) mentioned that patients treated presented several comorbidities. For the several comorbidities category, obesity was mentioned in combination with cardiovascular disease (twice) and hepatic disease (once). The remaining respondents in this category (n = 5) indicated that patients generally have various comorbidities but did not specify. Cardiovascular diseases were reported by 28% (n = 7) of respondents as common among their patients, followed by obesity at 24% (n = 6), and hepatic diseases at 16% (n = 4).

The respondents noted that medical treatment is often hindered by patients’ non-compliance, low levels of education, and low socioeconomic status, which also limits access to available treatment options.

#### 3.2.2. Distribution of Encountered DFU Types

The healthcare professionals who answered the questionnaire indicated that they handle various types of wounds, including neuropathic, ischemic, or neuroischemic wounds located at different pressure points on the foot. Each wound varies in depth, size, and infection status. Ten respondents said they often encounter wounds on patients’ toes. Eight respondents did not specify a location, seven found more on the plantar area of the foot, and four found them on the heels.

#### 3.2.3. Diabetic Foot Guidelines

Most of the respondents (59%, n = 17) reported following a published treatment guideline, while 31% (n = 9) indicated they did not follow any guidelines because of the heterogeneity of the patient population. The guidelines referenced by respondents include IWGDF, M.O.I.S.T. (moisture balance, oxygen balance, infection control supporting strategies, and tissue management) [46], T.I.M.E. (Tissue management, Infection and inflammation control, Moisture balance, and Edge of the wound) [47], the Diagnostic Therapeutic Care Path, ESVS Guidelines, the MD Approach, Best Practice Guidelines, and the DF Protocol, as well as various national and international guidelines.

#### 3.2.4. Infected DFUs

Most respondents (69%, n = 20) stated that they commonly encounter infected wounds, 17% (n = 5) said that the wounds they treat are usually not infected, 7% indicated that it is variable (n = 2), and 7% (n = 2) did not answer.

Fifteen distinct treatment options for infection were mentioned, and as respondents could answer more than once, a total of 45 answers were registered. Antibiotics were the most frequently mentioned intervention (31.1%, n = 14), followed by surgical debridement (26.7%, n = 12). Silver-based products (13.3%, n = 6) and antimicrobial dressings (6.7%, n = 3) were also reported. The remaining interventions (22.2%, n = 10) included a variety of approaches such as skin grafts, betadine, negative pressure wound therapy, honey, Granudacyn solution, topical oxygen therapy, and iodine.

#### 3.2.5. Wound Healing Time

Respondents mentioned that wound healing time depends on multiple factors as illustrated by the following quote:

“*Healing time is often related to the type of ulcer (…). The factors that most contribute to the delay in healing are infection, ischemia, compliance, and I would add the process of management.*”[P21, wound care specialist]

Some respondents provided an estimated healing time for the wounds they usually encounter; these values were divided into categories by the authors. In total, 31.0% (n = 9) of the respondents did not give an approximate healing time; they mentioned that it varies greatly from case to case. A total of 27.5% (n = 8) of respondents indicated that wounds took between 1 and 3 months to heal, and an additional 27.5% (n = 8) mentioned that wounds took more than 3 months to heal; most wounds taking more than 3 months to heal were specified to be neuroischemic. At the same time, 14.0% (n = 4) stated that the wounds they treat heal between 1 and 4 weeks.

#### 3.2.6. Current Treatment Techniques and Limitations

There is a considerable variation in the products used by medical professionals for the treatment of DFUs, with respondents mentioning a total of twenty-eight different products. Forty-eight answers were recorded, as respondents could provide more than one answer. Silver-based materials were the most mentioned by the respondents (28%, n = 8) in the treatment of infected wounds. Honey-based products were also commonly mentioned (17%, n = 5) for treating wounds with extensive necrotic tissue, as honey provides a wound healing environment.

To achieve non-load bearing on the wounds, respondents often used casting, noting that in such cases, additional wound care products need to be long-lasting (7 to 14 days) to minimize the frequency of cast removal. Respondents also pointed out that the high moisture levels commonly found in DFUs can make it challenging for patches like Urgotul to manage exudates effectively, requiring supplementary bandages. Limitations cited included the availability and affordability of specialized wound care products, as well as the fact that these specialized dressings are not always covered by insurance. Additionally, the need for tailored treatments and dressings at different stages of wound healing was seen as a drawback of current treatment options.

### 3.3. Section 2: Material Development

#### 3.3.1. Medical Professionals’ Perspectives on Ideal Treatment Features

According to the respondents, ideal features in material technologies for DFU treatment include the following:**Product application**: Medical professionals emphasize the importance of products that are easy to apply and manage, with some suggesting features like color change to indicate loss of functionality.**Treatment outcome**: Antibacterial activity is desired, with many respondents indicating a preference for materials that provide antibacterial effects.**Health economics**: Treatments should be affordable and ideally reimbursed by health insurance to ensure broader accessibility and adherence.**Healing time**: Products that expedite the healing process of DFUs are highly desired.**Additional properties**: Other desired properties include moisture control, non-adherence to the wound, breathability, and the ability to determine bacterial types present for more targeted treatment.

Figure 1 shows a more detailed overview of the ideal material technology characteristics mentioned by the respondents.

When asked which biological processes should be stimulated, respondents were able to provide more than one answer (see Table 1). They indicated that for effective wound healing, a product should primarily promote processes like angiogenesis (21%, n = 6) and granulation (24%, n = 7), along with re-epithelialization and fibroblast stimulation (14%, n = 4 for both of them). Additionally, 10% (n = 3) of the respondents mentioned that the product should stimulate every process involved in wound healing. Conversely, respondents said that the product should avoid causing adverse effects, particularly inflammation (28%, n = 8) and infection (14%, n = 4), which are critical to prevent. Additional concerns include avoiding the promotion of blood clots and necrosis (with 14% [n = 4] and 7% [n = 2], respectively), as well as minimizing pain and scar tissue formation, and ensuring that the product does not contribute to necrosis (7%, n = 2).

The desired duration of antibacterial properties includes a variety of responses. Most respondents indicated that antibacterial properties should last during the time the material is in direct contact with the wound.

#### 3.3.2. Frequency of Product Exchange

Most respondents (52%, n = 15) expressed they would use a product that required daily exchange if it resulted in faster wound healing. Meanwhile, 24% (n = 7) stated they would not use it, and 10% (n = 3) indicated they would use it depending on the circumstances.

The respondents who answered negatively often mentioned that it is too time-consuming and costly for them to make the change. Respondents who answered “maybe” said they would use the product if the patient could apply it properly and independently. However, they remained concerned about the potential challenges this might pose for patients.

*“(…) we need to consider that patients are usually of advanced age, with poor eyesight and reduced mobility. They will most likely not be able to reach the wound to treat it, they also usually live alone*.”[P4, Diabetic foot nurse]

#### 3.3.3. The Most Important Factor in the Wound Healing Process

When asked what they considered the most important factor in the wound healing process, one of the medical professionals said:

“*It’s not just the bandage that heals the patient, it’s the diet, physical activity, do they smoke or not? It is a combination of everything*.”[P1, Wound consultant]

The need for a holistic approach to wound healing is mentioned by eight of the respondents, as illustrated by the following quote:

“*The state of the patient, state of the wound, wound pre-treatment, person who treats the wound, material used to cover the wound. Each one of these are key factors in wound healing, together they all contribute to the healing process*.”[P22, wound care specialized nurse]

Educating the end users on how to apply the products and their working mechanisms is a recurring topic in the respondents’ answers (n = 6):

“*Education is key, know how to treat the wound and how the product works*.”[P1, Wound Consultant]

One of the respondents indicated that current products can be effective when used correctly.

## 4. Discussion

This study addressed three primary objectives: first, to inquire into the current challenges, limitations, and unmet user needs faced by healthcare professionals and material scientists in treating diabetic foot ulcers (DFUs); second, to inquire into what is considered an ideal treatment for DFUs by these stakeholders; and third, to create an overview of the key considerations for developing innovative therapeutic material technologies for DFU treatment. To achieve these objectives, an open-ended questionnaire was distributed through medical associations and conferences.

Understanding the types of wounds treated by respondents and the context of their patient populations is crucial for developing effective new products. Most respondents treat patients above 60 years of age who have a combination of comorbidities. Respondents also mentioned that these patients usually have lower levels of education, making it difficult for them to follow treatment plans, and they also have a lack of access to treatment options. Consequently, 24% (n = 7) of respondents indicated that patients might not be able to change a product by themselves daily and prefer longer-lasting options that a nurse could manage. Common wound locations mentioned by the respondents were the toes and the plantar area. This information is relevant to determine user and application needs for new material technologies in DFU treatment.

### 4.1. Product Characteristics

The ideal new product characteristics identified by respondents are consistent with Roger’s principles for the diffusion of innovations [48]. Ease of use was the most commonly mentioned feature, aligning with Rogers’ concept of minimizing complexity to facilitate adoption [48]. Respondents emphasized that products should be easily adaptable to various wound shapes and sizes, which increases compatibility with the existing practices and needs of healthcare professionals. This adaptability is crucial as it enhances the product’s fit within current workflows, thereby increasing the likelihood of clinical adoption.

Cost emerged as another major consideration for the respondents when developing new material technologies, highlighting the importance of relative advantage [48]; new products must be cost-effective to gain acceptance, especially among patients from lower socioeconomic backgrounds who may struggle with expensive treatments. Studies such as the one from Ka et al. (2022) [49] have pointed out that a low socioeconomic level is a barrier to patients’ adherence and access to medical treatments. The significance of relative advantage is further underlined by the need for insurance coverage, which can mitigate the financial burden on patients. To achieve this, the newly developed product must show clear evidence of superior efficacy compared to existing alternatives, enhancing its observability by making its benefits more apparent to users [48].

Respondents also pointed out the importance of educational materials accompanying the product, targeted at patients and healthcare professionals. Effective wound treatment options already exist, but improper use due to a lack of education can hinder positive outcomes. By providing these materials, the product enhances its trialability [48], allowing users to learn and experiment with its use in a controlled manner, thereby reducing perceived risk. Education is a well-established behavior change technique [50]; an example of this is an educational intervention carried out by Sikkens et al. (2018) [51]. They performed a study on antibiotic prescribing skills and showed that a targeted e-learning course led to an 11% higher exam pass rate among medical students compared to a control group, demonstrating the long-term effectiveness of educational interventions [51].

In summary, these characteristics—ease of use, adaptability, cost-effectiveness, demonstrable efficacy, and educational support—align with Rogers’ innovation characteristics and collectively improve the likelihood of successful product adoption.

### 4.2. Guidelines for DFU Treatment

There is no single agreed-upon way to treat DFUs, with respondents citing up to 10 available treatment guidelines. Interestingly, 35% (n = 10) of the respondents reported that they did not follow any guidelines and instead tailored treatment to individual patients’ needs and comorbidities due to the heterogeneity of patients. In this case, their experience plays a key role in choosing the best treatment.

Implementing DFU treatment guidelines is challenging due to various factors, including resource limitations, disparities in healthcare infrastructure, and the heterogeneity of patient needs. Variability in outcomes, even within the same country, can be attributed to differences in healthcare professionals’ training and emphasis on DFU management [45]. In low-resource settings, the limited availability of specialized care and resources further complicates adherence to guidelines [52]. Additionally, DFU treatment often requires a customized approach, depending on factors like ulcer severity, patient comorbidities, and access to offloading devices or surgical options.

Given these challenges, healthcare providers may not strictly adhere to guidelines, opting instead to rely on their clinical experience and judgment. While personalizing care is important, established guidelines based on scientific research [53] advocate for a team-based approach to patient care [54], which can improve outcomes, prevent amputations, and reduce the overall impact of diabetes-related foot disease [25]. This highlights the complex landscape of DFU treatment, where both adherence to guidelines and individual professional experience play crucial roles.

### 4.3. Infection in DFU

Infections in diabetic foot ulcers (DFUs) are complex [55] and multifaceted. An infected wound in the context of DFUs is generally identified by the presence of clinical symptoms such as redness, warmth, swelling, pain, or discharge [56]. These symptoms can be subtle in diabetic patients due to neuropathy and poor blood circulation [57]. The timing of intervention is particularly challenging, as overtreatment with antibiotics can lead to antimicrobial resistance [58], while delayed treatment of true infections can result in serious complications, including limb loss [59]. Current guidelines recommend initiating treatment when clear signs of infection are present and discontinuing antibiotics when signs of infection have resolved [58], emphasizing the importance of individualized patient assessment.

Reflecting this challenge, 74% (n = 21) of respondents reported that the wounds they treat are often infected and mentioned antibiotics as the most common treatment for infection. Nearly all respondents (93%, n = 27) expressed a desire for products with antibacterial properties, and 33% out of this 93% (n = 9) wanted the antibacterial properties to last as long as the product was in contact with the wound.

Diabetic foot infections (DFIs) affect 50–60% of patients with DFUs [60]. A common pathogen in DFIs is *Staphylococcus aureus*, comprising 20–25% of bacteria isolated from DFIs [61]. *S. aureus* has shown resistance to several antibiotics, including methicillin (MRSA) [62], making it difficult to treat. This suggests a need for wound care products with antibacterial properties that do not rely solely on traditional antibiotics. These products could complement current treatments by having a different mode of action to kill bacteria compared to antibiotics in combating infection.

Debridement, which removes necrotic and avascular tissue [63], was the second most mentioned treatment for infected DFUs. This is a crucial step in treatment since the first stage of a diabetic foot infection by *S. aureus* is the bacteria attaching to surface components of the skin, such as epidermal keratinocytes [64]. Since there are already good methods for debridement, this should not be the main focus in the development of new material technologies for DFUs.

Silver-based products have been used as a complementary treatment for infection in diabetic foot ulcers. This was the third most mentioned treatment by the respondents in our questionnaire. However, the widespread use of silver-containing compounds has led to the emergence of silver-resistant bacteria, including four different *S. aureus* isolates in the study by Hosny et al. (2019) [65]. The potential for resistance development may depend on the specific silver formulation and the exposure conditions. These factors should be taken into consideration during product development.

### 4.4. Biological Processes in Wound Healing

The respondents indicated specific biological processes that new products should or should not stimulate. Angiogenesis was the most frequently mentioned process that respondents desired the product to stimulate. We understand that angiogenesis is essential for proper wound healing. However, it is important to note that excessive angiogenesis does not necessarily guarantee faster healing and may even have negative consequences [66]. The level of angiogenesis often coincides with the inflammatory response because inflammatory cells produce numerous factors that promote angiogenesis [67]. While inflammation is a necessary part of healing, excessive inflammation can lead to poor healing outcomes and increased scarring [68]. For this reason, it is crucial to carefully tune the material’s characteristics. While a few respondents indicated that, ideally, a product should stimulate all processes involved in wound healing, achieving this across all stages of wound healing with a single product is quite challenging. This is one of the reasons why it is important to consider new products and material technologies as pieces of a larger puzzle and to identify other material technologies alongside which they might work well. By identifying complementary products, we can explore the potential benefits of a combination therapy approach.

### 4.5. Strengths and Limitations

Open-ended questionnaires allow healthcare professionals to freely share their experiences, providing valuable insights into current DFU treatments and their shortcomings. Unlike fixed-choice formats, this approach prevents potential misleading answers and ensures we capture all relevant information. These detailed responses will inform the development of products that align closely with clinicians’ needs, are user-friendly, and address existing treatment limitations.

This study acknowledges limitations due to sample size, self-selection bias, and challenges with online surveys. A small sample size and self-selection bias prevent findings from being extrapolated as well as increase the possibility of assuming a false premise as true [69,70]. Finally, the online format makes it difficult to clarify responses, especially for open-ended questions. This can leave unclear or ambiguous data that require extra effort to interpret. While the sample size limits applying the findings to the entire continent, the results still provide valuable insights. They highlight the complex nature of DFU treatment and the importance of a holistic approach.

### 4.6. Design Recommendations and Future Directions

This paragraph aims to recapitulate the results from our three research objectives, emphasizing the need for material technology innovations in DFU treatment. Recognizing that a single product cannot address all aspects of DFU care, the development of new material technologies should focus on combination therapy and specific use cases. For example, products intended for use alongside offloading therapy should be durable enough to minimize disruptions caused by frequent device removal. The ideal new product for DFU treatment should prioritize features like ease of use, enhanced antibacterial properties for infected wounds, affordability, and targeted biological activity, such as promoting angiogenesis without triggering excessive inflammation. By aligning new materials with these desired characteristics, we can effectively address the limitations of current treatments, leading to improved management of DFUs and, ultimately, better overall patient care. Future research should also explore variations in treatment practices globally, given the differences in healthcare resources [71] and patient demographics.

## 5. Conclusions

Healthcare professionals pointed out several limitations in current material technologies for the treatment of DFUs, such as the limited availability and high cost of specialized wound care products as well as the need for various treatments and dressings to cover all stages of wound healing. When developing new material technologies, it is crucial to align with the specific needs of users of healthcare professionals. According to healthcare professionals, an ideal product for the treatment of DFUs should feature products that offer long-lasting antibacterial properties, are easy to use, and are cost-effective. Additionally, the education of healthcare professionals and patients emerged as a critical theme, emphasizing the importance of proper technology application and education for optimal treatment results in DFUs.

This study highlights the complex nature of treating diabetic foot ulcers (DFUs). The survey responses suggest that a single, universal treatment solution is unlikely to meet the diverse needs of DFU management. Therefore, when designing new products, developers should not only consider user needs but also how their material technology can work in conjunction with complementary wound healing products.

## Figures and Tables

**Figure 1 biomedicines-12-02483-f001:**
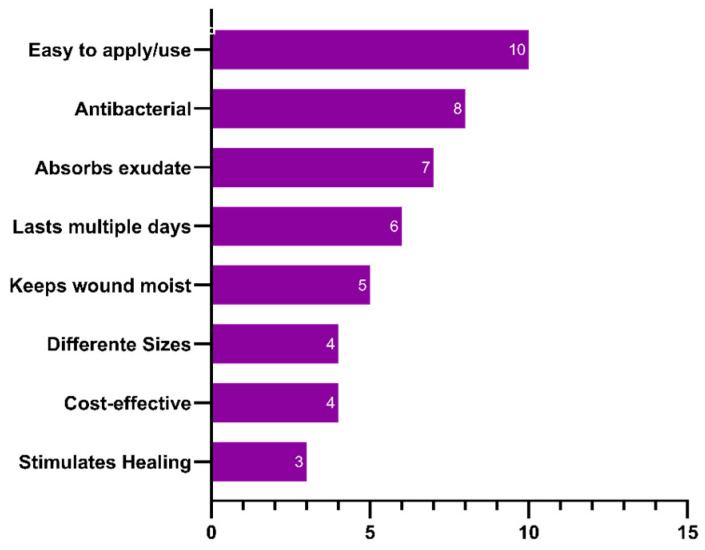
Frequency of desired product characteristics reported by respondents. Data from 29 respondents. Multiple responses were allowed (47 registered responses).

**Table 1 biomedicines-12-02483-t001:** Biological processes that are considered essential and detrimental to achieving wound healing according to the respondents.

Essential Product Characteristics	Counts	Detrimental Product Characteristics	Counts
Angiogenesis	6	Inflammation (immune response)	8
Fibroblast stimulation	4	Pain to the patient	1
Granulation	7	Scar tissue	1
Re-epithelialization	4	Blood clots	4
Reduction in inflammation	2	Infection	4
Every process involved in wound healing	3	Necrosis	2

## Data Availability

The data retrieved from the user needs questionnaire can be obtained through the following link: https://docs.google.com/spreadsheets/d/1wa4NG0h83wmD5T8WNYUkZb8r5XoVTAI9/edit?usp=sharing&ouid=110983971402187884564&rtpof=true&sd=true (accessed on 22 August 2024).

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
