# Peer review of "Material Technologies for Improved Diabetic Foot Ulcer (DFU) Treatment: A Questionnaire Study of Healthcare Professionals’ Needs"

_biomedicines, 2024, doi:10.3390/biomedicines12112483_

Round 1

Reviewer 1 Report

Comments and Suggestions for Authors

Dear Author, you submitted the manuscript titled "Material Technologies for Improved Diabetic Foot Ulcer (DFU) 2 Treatment: A Questionnaire Study of Healthcare Professionals’ 3 Needs" It has questionnaire-based studies of critical feedback and novel work highlighting the end product's importance and actual feeling. Despite the above work, I feel some minor comments or suggestions as follows

1. What is the sample size of your studies, and how do you calculate theoretically?

2. If some respondents have given negative or wrong feedback, what are the selection criteria?

3. Is there any ethical approval required for collecting data and publication with consent from the respondent?

ok

Author Response

Comment 1: What is the sample size of the study? How do you calculate it theoretically?

Response 1: The study included 29 participants. No initial sample size calculation was conducted. In qualitative research the concept of information power is used, this indicates that  the more information the sample holds, relevant for the actual study, the lower number of participants is needed [1]. We considered the sample sufficient once we reached data saturation, meaning no new themes emerged from the responses [2]. Given the highly specialized nature of the questionnaire and the target group the responses collected were determined adequate to meet our research objectives.

Comment 2: If some respondents have given negative or wrong feedback, what are the selection criteria?

Response 2: The questionnaire was designed to gather honest opinions from medical professionals, so all feedback is valuable. There is no "negative" or "wrong" feedback, as responses reflect the real experiences of those treating diabetic foot ulcers (DFU). Whether respondents see no need for improvements or suggest changes, their input helps us understand the current state of treatment and potential areas for development. No opinions were excluded from the analysis.

Comment 3: Is there any ethical approval required for collecting data and publication with consent from the respondents?

Response 3: The questionnaire did not collect specific patient data. Instead, it focused on the general experiences of healthcare professionals in treating diabetic foot ulcers (DFU) and their preferences for future treatments. This means that the opinions are professional opinions on material user needs in the context of their profession and not personal opinions.

Before completing the questionnaire, all respondents were informed that their responses would be used as input for a PhD project aimed at developing new material solutions for DFU treatment.

References

  1. Malterud, K.; Siersma, V.D.; Guassora, A.D. Sample Size in Qualitative Interview Studies. Qualitative Health Research 2016, 26, 1753-1760, doi:10.1177/1049732315617444.
  2. Sharma, S.K.; Mudgal, S.K.; Gaur, R.; Chaturvedi, J.; Rulaniya, S.; Sharma, P. Navigating Sample Size Estimation for Qualitative Research. Journal of Medical Evidence 2024, 5, 133-139, doi:10.4103/jme.Jme_59_24.

Reviewer 2 Report

Comments and Suggestions for Authors

Please find attached

Comments on the manuscript “Material Technologies for Improved Diabetic Foot Ulcer (DFU) Treatment: A Questionnaire Study of Healthcare Professionals’ Needs” submitted to Biomedicines MDPI for consideration.

Major concern: I feel that this paper is more relevant to journals majorly focusing on clinical data.

Minor comments.

1.     Line No. 17, replace ‘diabetes’ with ‘diabetic patients.’

2.     Line No. 32, combination therapies and specific use cases, the statement looks contradictory.

3.     The introduction looks fine, but the authors should highlight the rationale of this work.

4.     Is there any need of ethical statement or statement of consent for this research?

5.     Which major comorbidities complicate the DFU?

6.     Line No. 218, “Healing time is often related to the type of ulcer (…).” What does the dots mean in parentheses.

7.     The wound healing time is not mentioned clearly.

8.     The Figures or Tables cited in text should be either all bold or normal.

9.     Why in some places in the main document, there are dots within the parentheses?

10.  The Conclusions is not to the point. The authors should highlight the major findings within this section.

Comments on the Quality of English Language

Minor changes only

Author Response

Comment Major concern: I feel that this paper is more relevant to journals majority focusing on clinical data.

Response Major concern: This article aims to capture the clinical experiences of healthcare professionals and use their insights to guide the development of new novel material technologies for the use of diabetic foot ulcer (DFU) treatment. We are not presenting clinical data but want to identify real-world challenges and opportunities that could be addressed through new treatment solutions and that will steer new technology development based on user needs.

Minor concerns

Comment 1: Line 17: replace diabetes with diabetic patients

Response 1: Replacement was made.

Comment 2: Line 32: combination therapies and specific use cases, the statement looks contradictory.

Response 2: In line 32 “should” was replaced by “could” to make a less strong statement.

Ideally, we can find one material that suits all needs, but we do not expect this. Targeting all stages of wound healing and all different types of wounds, combined with infection eradication in a single material seems unrealistic. This is why we consider that treatments could be targeted to a specific type of wound and/or specific stage of wound healing and combined with other products that take care of the remaining biological processes involved.

Comment 3:  The introduction looks fine but the authors must highlight the rationale of this work

Response 3:  We agree, the following changes were made to highlight the rationale of this work.

Lines 85-89 indicate an increased challenge in treating DFU compared to other wounds.

Lines: 105 to 110: “Given the varied advantages and limitations of wound dressings, we were determined to investigate the perception of those who use advanced dressings for the treatment of DFUs. This prompted us to conduct a user needs questionnaire to determine what an ideal product for the treatment of DFUs would look like for healthcare professionals who work with DFUs”, were replaced by: “Given the advantages and limitations of current wound dressings, we decided to explore the user needs of healthcare professionals who use advanced dressings to treat diabetic foot ulcers (DFUs). The poor clinical outcomes of diabetic ulcers in comparison to non-diabetic ulcers [1,2], made it clear that this investigation was necessary. This led us to create a user needs questionnaire to investigate what an ideal product for treating DFUs would resemble from the perspective of healthcare professionals directly involved in such treatments.”

Comment 4: Is there any need of an ethical statement or statement of consent for this research?

Response 4: The questionnaire did not collect specific patient data. Instead, it focused on the general experiences of healthcare professionals in treating diabetic foot ulcers (DFU) and their preferences for future treatments. This means that the opinions are professional opinions on material user needs in the context of their profession and not personal opinions.

Before completing the questionnaire, all respondents were informed that their responses would be used as input for a PhD project aimed at developing new material solutions for DFU treatment.

Comment 5: Which major comorbidities complicate the DFU?

Response 5:  Cardiovascular, renal, and neurological diseases are among the most common comorbidities that complicate the treatment of DFUs.

Peripheral arterial disease (PAD) is one of the main causes of diabetic foot problems along with neuropathy. PAD causes narrowing of the arteries [3]. Reduced blood flow creates an unfavorable environment for wound healing, as tissues require adequate perfusion to repair and regenerate.

Chronic kidney disease (CKD) is associated with prolonged healing times for DFUs and a higher rate of non-healing ulcers. Patients with renal disease, especially those on dialysis, experience poorer DFU outcomes compared to those with normal kidney function [4].

Line 42-43: Mentions the impact of neuropathy in DFUs

In Lines 54-55, PAD is mentioned as one of the contributing factors in DFUs.

Chronic kidney disease was added to line 56

Comment 6: Line 218: “Healing time is often related to the type of ulcer (…)”What do the dots in parenthesis mean?

Response 6: The ellipses in parentheses indicate that the quote is longer, but it was shortened. https://www.uefap.com/writing/function/int.htm

Comment 7: The wound healing time is not mentioned clearly

Response 7: The wound healing times are presented in intervals, as responses varied in the exact number of days or weeks. Grouping the data into clusters allowed for a clearer presentation. If the reviewer prefers, we can provide the specific time points mentioned by the respondents in an online figure or supplementary information.

Comment 8: Figures or tables cited in the text should be all bold or normal

Response 8: All figures and tables cited in the text were formatted in bold. (Lines 263, 266, 269 and 279)

Comment 9: Why in some places of the main document there are dots between parenthesis?

Response 9: 

The ellipses in parentheses indicate that the quote is longer, but it was shortened. https://www.uefap.com/writing/function/int.htm

Comment 10: The conclusion is not to the point. The authors should highlight major findings within this section.

Response 10: We agree, the conclusion was modified to highlight major findings and present actionable recommendations.

Previous conclusion:

This study reinforces the complex nature of DFUs treatment. The survey response indicates that a single, universal treatment solution is unlikely. When developing new material technologies, adherence to the user needs is crucial. Furthermore, the education of healthcare professionals and patients emerged as a critical theme, emphasizing the importance of proper technology application and education for optimal treatment results in DFUs.”

Revised conclusion: (Lines 462-477) page 10.

Healthcare professionals pointed out several limitations in current material technologies for the treatment of DFUs, such as the limited availability and high cost of specialized wound care products as well as the need for various treatments and dressings to cover all stages of wound healing. When developing new material technologies, it is crucial to align with the specific user needs of healthcare professionals. According to healthcare professionals, an ideal product for the treatment of DFU should feature products that offer long-lasting antibacterial properties, are easy to use, and are cost-effective. Additionally, the education of healthcare professionals and patients emerged as a critical theme, emphasizing the importance of proper technology application and education for optimal treatment results in DFUs.

This study highlights the complex nature of treating diabetic foot ulcers (DFUs). The survey responses suggest that a single, universal treatment solution is unlikely to meet the diverse needs of DFU management. Therefore, when designing new products, developers should not only consider user needs but also how their material technology can work in conjunction with complementary wound-healing products.

References:

  1. Jeffcoate, W.J.; Harding, K.G. Diabetic foot ulcers. Lancet 2003, 361, 1545-1551, doi:10.1016/S0140-6736(03)13169-8.
  2. Armstrong, D.G.; Boulton, A.J.M.; Bus, S.A. Diabetic Foot Ulcers and Their Recurrence. N Engl J Med 2017, 376, 2367-2375, doi:10.1056/NEJMra1615439.
  3. Tachi, K.; Gonda, K.; Kochi, T.; Niwa, J. Do peripheral arterial disease and smoking impede diabetic ulcer healing? Wounds International 2024, 15, 18-23.
  4. Bonnet, J.-B.; Sultan, A. Narrative Review of the Relationship Between CKD and Diabetic Foot Ulcer. Kidney International Reports 2022, 7, 381-388, doi:10.1016/j.ekir.2021.12.018.

Reviewer 3 Report

Comments and Suggestions for Authors

The manuscript “Material Technologies for Improved Diabetic Foot Ulcer (DFU) Treatment: A Questionnaire Study of Healthcare Professionals’ Needs“ by Guerrero et al. reports understand the user needs of healthcare professionals treating diabetic foot ulcers for newly developed material technologies. Although, an open-ended questionnaire was used to identify user needs, identify the limitations of current treatments, and determine the specific requirements for ideal treatment; the material technologies or the review of current biomedical materials seems lack from this work. Therefore, I would suggest authors may take at least a major revision. Here are the comments and suggestions:

1.     Previous studies to repones these questionnaires should be reviewed and tabled.

2.     What materials or their technologies are involved for the DFU treatments?

3.     Some current commercial products should be added and compared.

4.     The conclusions should be extended.

Author Response

Comment1: Previous studies to respond to this questionnaire should be reviewed and tabled

Comment 2: What materials or technologies are involved in the treatment of DFU?

Comment 3: Some current commercial products should be added and compared.

Response to comments 1, 2 and 3

We conducted a systematic review of materials developed to be used for the treatment of DFU [1]. This review was used to inform the development of the user needs questionnaire, identifying key characteristics considered by material developers in the design of material technologies for diabetic foot ulcer (DFU) treatment.

The goal of applying a user needs questionnaire to healthcare professionals is to explore what innovations could be developed to better meet clinical needs in the treatment of DFU. While we understand the importance of comparing ideal product characteristics with those of existing products, we believe this would detract from our primary objective, which is to guide future material development based on real-world clinical experience.

Comment 4: The conclusions should be extended.

Response 4: 

The conclusion was modified to highlight major findings and present actionable recommendations.

Previous conclusion:

This study reinforces the complex nature of DFUs treatment. The survey response indicates that a single, universal treatment solution is unlikely. When developing new material technologies, adherence to the user needs is crucial. Furthermore, the education of healthcare professionals and patients emerged as a critical theme, emphasizing the importance of proper technology application and education for optimal treatment results in DFUs.”

 Revised conclusion:

(Lines 462-477) page 10.

Healthcare professionals pointed out several limitations in current material technologies for the treatment of DFUs, such as the limited availability and high cost of specialized wound care products as well as the need for various treatments and dressings to cover all stages of wound healing. When developing new material technologies, it is crucial to align with the specific user needs of healthcare professionals. According to healthcare professionals, an ideal product for the treatment of DFU should feature products that offer long-lasting antibacterial properties, are easy to use, and are cost-effective. Additionally, the education of healthcare professionals and patients emerged as a critical theme, emphasizing the importance of proper technology application and education for optimal treatment results in DFUs.

This study highlights the complex nature of treating diabetic foot ulcers (DFUs). The survey responses suggest that a single, universal treatment solution is unlikely to meet the diverse needs of DFU management. Therefore, when designing new products, developers should not only consider user needs but also how their material technology can work in conjunction with complementary wound-healing products.

References

  1. Vargas Guerrero, M.; Aendekerk, F.M.A.; de Boer, C.; Geurts, J.; Lucchesi, J.; Arts, J.J.C. Bioactive-Glass-Based Materials with Possible Application in Diabetic Wound Healing: A Systematic Review. Int J Mol Sci 2024, 25, doi:10.3390/ijms25021152.

Reviewer 4 Report

Comments and Suggestions for Authors

This paper explores healthcare professionals' user needs for diabetic foot ulcer (DFU) treatments, focusing on the development of new material technologies for the improvement of DFU care. In this sense, due to the interest of the topic that the work addresses, I find it of utility for the scientific community. Therefore, I think that it could be suitable for publication in the Biomedicines journal provided that the following comments are implemented within the document: 

- In order to emphasize the complexity of DFU care, please address how DFU treatments may vary depending on particular patient needs.

- How economic-type barriers may affect DFU treatment in different regions? Please comment.

- The authors could include some research on successful joint therapies in order to inform future material technologies.

- Comment on how patient lifestyle could be integrated into material design.

- Please address successful educational interventions from other healthcare fields in order to illustrate the importance of such role.

- How new materials could lower treatment costs? Please comment.

- Please include in the Conclusions section actionable final recommendations.

Comments on the Quality of English Language

-

Author Response

Comment 1: In order to emphasize the complexity of DFU care, please address how DFU treatments may vary depending on particular patient needs.

Response 1: We understand how challenging it is to treat DFU and that treatments will vary depending on the patient's needs. However, we consider that addressing specific treatment scenarios will make us lose focus. For example, Offloading in adequate shoes might be enough for certain ulcers whereas in other cases complex surgery might be necessary to prevent amputation. Treatment trajectories are often time-consuming and conflict with the patient’s lack of understanding of the disease and risks.

Lines 72 to 74 were modified to emphasize the importance of individualized treatment.

Previous sentences: The International Working Group on the Diabetic Foot (IWGDF) Guidelines recommends that DFU treatment should involve a multidisciplinary team.

Revised sentence:  The International Working Group on the Diabetic Foot (IWGDF) Guidelines recommends that DFU treatment should involve a multidisciplinary team to provide individualized care tailored to the complex needs of each patient [1].

Comment 2: How economic-type barriers may affect DFI treatment in different regions? Please comment.

Response 2: Prolonged hospitalization and long-lasting antibiotic treatments along with complex surgery might not be available in all countries. Health insurance costs might not be accessible depending on the country.

Developing countries usually struggle with limited specialized personnel to treat the complexity of DFU and DFI, limited availability of specialized wound care products and inadequate healthcare infrastructure, particularly in rural areas[1]

Up next we illustrate our point with some examples:

  • The limited amount of economic resources limits the treatment possibilities, a study on barriers to diabetic foot care performed in Barbados mentions that at the time of the study, there were only 3 podiatrists on the island. In Barbados 14.6 % of the population aged 20 to 79 is estimated to have diabetes, just under 30 000 people, meaning at that time there was only 1 podiatrist per 10 000 patients [2].
  • In Italy, southern and central areas generally face more barriers in both hospital and community settings compared to northern regions. Northern Italian regions generally reported fewer barriers, possibly due to a higher concentration of specialized diabetic foot centers [3]. It is relevant to point out that Northern Italy has a higher economic output compared to the South, In 2019, the per capita GDP of Lombardy (a northern region) was €39 700, while Calabria (a southern region) had a per capita GDP of only €17 300 [4].

Comment 3:  The authors could include some research on successful join therapies in order to inform future material technologies

Response 3:  Our goal when applying a user needs questionnaire to healthcare professionals was to explore what innovations could be developed to better meet clinical needs in the treatment of DFU. While we understand the importance of showing success stories of join therapies, we consider this would detract from our primary objective, which is to guide future material development based on real-world clinical experience.

Comment 4: Comment on how patient lifestyle could be integrated into material design

Response 4:  Materials for the treatment of DFUs should ideally adapt to a patient's lifestyle, but certain habits, like smoking, and unhealthy diets require change on the patient's part. For active individuals, healing—such as for a foot wound—may need adjustments to their physical activities. While the goal is for patients to continue their daily routines with minimal disruption, compromises are inevitable. The material should prioritize patient comfort without sacrificing its effectiveness. Ideally, it would support weight-bearing, be self-applicable, and allow the patient to maintain their normal life as much as possible during recovery.

Comment 5:  Please address successful educational interventions from other healthcare fields in order to illustrate the importance of such role.

Response 5:

Lines 362-366 were added to illustrate the importance of educational interventions. “Education is a well-established behavior change technique [5], an example of this is an educational intervention done by Sikkens et al.(2018)[6]. They performed a study on antibiotic prescribing skills and showed that a targeted e-learning course led to an 11% higher exam pass rate among medical students compared to a control group, demonstrating the long-term effectiveness of educational interventions[6].”

To highlight the significance of educational interventions for the use of therapeutic materials in healthcare, we can draw from successful examples in other healthcare fields. Education is a well-established behavior change technique [5]. For instance, a study examining antibiotic prescribing skills among medical students demonstrated the long-term impact of e-learning. Six months after the students took a 6-week course focusing on antibiotic prescribing skills, students who underwent the training outperformed those who did not, with 11% more students passing the exam in the trained group compared to the control group [6]. The study concludes that E-learning during a limited period can significantly improve medical students’ performance of an antimicrobial therapeutic consultation in a situation simulating clinical practice 6 months later. This result demonstrates how targeted educational interventions can lead to measurable improvements in healthcare competencies, ultimately enhancing patient care outcomes.

DeMarco et al.[7] highlight the importance of educational interventions to empower patients with knowledge about their conditions, which leads to better adherence to treatment plans, lifestyle changes, and disease management. Patient education reduces hospital readmission rates by ensuring patients understand their discharge instructions and how to manage their health at home. Educational materials improve doctor-patient relationships, boost patient satisfaction, and are crucial for promoting long-term positive health behaviors​

These examples illustrate the importance of education in healthcare, for future medical professionals and patients.

Comment 6: How new materials could lower treatment costs? Please comment

Response 6:  New materials could potentially reduce treatment costs by accelerating wound healing and improving healing rates. Faster recovery would lower healthcare expenses for patients and reduce the overall societal burden, as patients could return to work sooner. However, this is an educated assumption, as we do not yet have data to confirm these cost reductions, since the products have not been developed.

Comment 7: Please include in the conclusions section actionable final recommendations

Response 7:  The conclusion was modified to highlight major findings and present actionable recommendations.

Previous conclusion:

This study reinforces the complex nature of DFUs treatment. The survey response indicates that a single, universal treatment solution is unlikely. When developing new material technologies, adherence to the user needs is crucial. Furthermore, the education of healthcare professionals and patients emerged as a critical theme, emphasizing the importance of proper technology application and education for optimal treatment results in DFUs.”

Revised conclusion:

(Lines 473-488) page 10.

Healthcare professionals pointed out several limitations in current material technologies for the treatment of DFUs, such as the limited availability and high cost of specialized wound care products as well as the need for various treatments and dressings to cover all stages of wound healing. When developing new material technologies, it is crucial to align with the specific user needs of healthcare professionals. According to healthcare professionals, an ideal product for the treatment of DFU should feature products that offer long-lasting antibacterial properties, are easy to use, and are cost-effective. Additionally, the education of healthcare professionals and patients emerged as a critical theme, emphasizing the importance of proper technology application and education for optimal treatment results in DFUs.

This study highlights the complex nature of treating diabetic foot ulcers (DFUs). The survey responses suggest that a single, universal treatment solution is unlikely to meet the diverse needs of DFU management. Therefore, when designing new products, developers should not only consider user needs but also how their material technology can work in conjunction with complementary wound-healing products.

References

  1. Swaminathan, N.; Awuah, W.A.; Bharadwaj, H.R.; Roy, S.; Ferreira, T.; Adebusoye, F.T.; Ismail, I.F.N.b.; Azeem, S.; Abdul‐Rahman, T.; Papadakis, M. Early intervention and care for Diabetic Foot Ulcers in Low and Middle Income Countries: Addressing challenges and exploring future strategies: A narrative review. Health Science Reports 2024, 7, doi:10.1002/hsr2.2075.
  2. Guell, C.; Unwin, N. Barriers to diabetic foot care in a developing country with a high incidence of diabetes related amputations: an exploratory qualitative interview study. BMC Health Services Research 2015, 15, doi:10.1186/s12913-015-1043-5.
  3. Meloni, M.; Acquati, S.; Licciardello, C.; Ludovico, O.; Sepe, M.; Vermigli, C.; Da Ros, R. Barriers to diabetic foot management in Italy: A multicentre survey in diabetic foot centres of the Diabetic Foot Study Group of the Italian Society of Diabetes (SID) and Association of Medical Diabetologists (AMD). Nutrition, Metabolism and Cardiovascular Diseases 2021, 31, 776-781, doi:10.1016/j.numecd.2020.10.010.
  4. Fernández Villaverde, J.; Laudati, D.; Ohanian, L.; Quadrini, V. Accounting for the duality of the Italian economy. Available online: https://cepr.org/voxeu/columns/accounting-duality-italian-economy (accessed on
  5. Kok, G.; Gottlieb, N.H.; Peters, G.-J.Y.; Mullen, P.D.; Parcel, G.S.; Ruiter, R.A.C.; Fernández, M.E.; Markham, C.; Bartholomew, L.K. A taxonomy of behaviour change methods: an Intervention Mapping approach. Health Psychology Review 2015, 10, 297-312, doi:10.1080/17437199.2015.1077155.
  6. Sikkens, J.J.; Caris, M.G.; Schutte, T.; Kramer, M.H.H.; Tichelaar, J.; van Agtmael, M.A. Improving antibiotic prescribing skills in medical students: the effect of e-learning after 6 months. Journal of Antimicrobial Chemotherapy 2018, 73, 2243-2246, doi:10.1093/jac/dky163.
  7. DeMarco, J.; Nystrom, M.; Salvatore, K. The Importance of Patient Education Throughout the Continuum of Health Care. Journal of Consumer Health On the Internet 2011, 15, 22-31, doi:10.1080/15398285.2011.547069.

Round 2

Reviewer 3 Report

Comments and Suggestions for Authors

The title is incorrect; and it's more like "Products" but not "Material Technologies" for Improved Diabetic Foot Ulcer (DFU) Treatment